# Hypoxia Rapidly Induces the Expression of Cardiomyogenic Factors in Human Adipose-Derived Adherent Stromal Cells

**DOI:** 10.3390/jcm8081231

**Published:** 2019-08-15

**Authors:** Jung-Won Choi, Hanbyeol Moon, Seung Eun Jung, Soyeon Lim, Seahyoung Lee, Il-Kwon Kim, Hoon-Bum Lee, Jiyun Lee, Byeong-Wook Song, Sang Woo Kim, Ki-Chul Hwang

**Affiliations:** 1Institute for Bio-Medical Convergence, College of Medicine, Catholic Kwandong University, Gangneung-si, Gangwon-do 210-701, Korea; 2Department of Integrated Omics for Biomedical Sciences, Graduate School, Yonsei University, Seoul 03722, Korea; 3International St. Mary’s Hospital, Catholic Kwandong University, Incheon Metropolitan City 22711, Korea; 4Department of Plastic and Reconstructive Surgery, International St. Mary’s Hospital, Catholic Kwandong University, Incheon Metropolitan City 22711, Korea; 5Department of Medical Science, College of Medicine, Catholic Kwandong University, Gangneung-si, Gangwon-do 25601, Korea

**Keywords:** adipose-derived adherent stromal cells (ADASs), adipose-derived stem cells (ASCs), cardiomyocyte differentiation, hypoxia, stromal vascular fraction (SVF)

## Abstract

Background: The efficacy of interstitial vascular fraction (SVF) transplantation in the treatment of heart disease has been proven in a variety of in vivo studies. In a previous study, we found that bone marrow-derived mesenchymal stem cells (BM-MSCs) altered their expression of several cardiomyogenic factors under hypoxic conditions. Methods: We hypothesized that hypoxia may also induce obtained adipose-derived adherent stromal cells (ADASs) from SVFs and adipose-derived stem cells (ASCs) to differentiate into cardiomyocytes and/or cells with comparable phenotypes. We examined the differentiation markers of cell lineages in ADASs and ASCs according to time by hypoxic stress and found that only ADASs expressed cardiomyogenic markers within 24 h under hypoxic conditions in association with the expression of hypoxia-inducible factor 1-α (HIF-1α). Results: Differentially secreted proteins in a conditioned medium (CM) from ASCs and ADASs under normoxic or hypoxic conditions were detected using an antibody assay and may be associated with a dramatic increase in the expression of cardiomyogenic markers in only ADASs. Furthermore, the cardiomyogenic factors were expressed more rapidly in ADASs than in ASCs under hypoxic conditions in association with the expression of HIF-1α, and angiogenin, fibroblast growth factor-19 (FGF-19) and/or macrophage inhibitory factor (MIF) are related. Conclusions: These results provide new insights into the applicability of ADASs preconditioned by hypoxic stress in cardiac diseases.

## 1. Introduction

Purified aqueous stromal vascular fractions (SVFs) generated by the enzymatic digestion of lipoaspirate have been widely studied for their regenerative potential [1,2,3,4,5]. This aqueous fraction includes adipose-derived stem cells (ASCs), endothelial (progenitor) cells, immune cells, fibroblasts, smooth muscle cells, pericytes and other stromal components as a heterogeneous cell population [6]. The regenerative effects of SVF can be caused by the heterogeneity of the cell population, which likely contributes numerous mechanisms to facilitate regeneration [5]. The regenerative capabilities of SVFs and/or ASCs have been observed and applied in diverse animal experiments and clinical trials [4], such as in studies of multiple sclerosis [7], diabetes [8], burn injuries [9], Crohn’s disease [10], bone disease [11] and cardiac defects [2,12,13]. Various in vivo studies have demonstrated the efficacy of ASC implantation in the treatment of acute myocardial infarction [14], ischemic cardiomyopathy [15], dilated cardiomyopathy [16], hindlimb ischemia [17] and stroke [18]. SVFs containing ASCs have the potential to regenerate the cardiovascular system via direct differentiation into vascular endothelial cells (VECs), vascular smooth muscle cells (VSMCs) and cardiomyocytes, fusion with tissue-resident cells, and the production of paracrine factors [2]. In addition to the potent regenerative effects of SVFs in many defects, this strategy has attracted attention because adipose tissues are abundant, can be easily collected using liposuction with minimal donor inconvenience and contain a significantly higher proportion of stem cells than bone marrow [2].

In a previous study, we found that bone marrow-derived mesenchymal stem cells (BM-MSCs) altered the expression of several cardiomyocyte differentiation-related factors under hypoxic conditions and provided new insights into the applicability of MSCs preconditioned by hypoxic stimulation for use in cardiac diseases [19]. Based on our previous study, we hypothesized that hypoxia may also induce adipose-derived adherent stromal cells (ADASs) from SVFs and ASCs to differentiate into cardiomyocytes and/or cells with comparable phenotypes. In this study, we investigated the expression of cell cardiomyocyte differentiation markers in ADASs/ASCs by hypoxia and found that only ADASs expressed cardiomyocyte differentiation markers within 24 h in the hypoxic state in association with the expression of hypoxia-inducible factor 1-α (HIF-1α). In addition, differentially secreted proteins in a conditioned medium (CM) of ASCs and ADASs under normoxic or hypoxic conditions were detected using an antibody assay, suggested these proteins may be associated with the rapid increase in the expression of cardiomyogenic markers in ADASs.

## 2. Experimental Section

### 2.1. Adipose Tissue Harvesting, Isolation, and Culture of ADASs and ASCs

Human adipose tissue samples for ADAS isolation were obtained from four donors (men in their 40s to 60s) who were recruited at International St. Mary’s Hospital of the Catholic Kwandong University. The study protocol was approved by the ethics review committee of the Institutional Review Board (IRB no. IS15OISE0023, approvedon 1 April 2015), College of Medicine, Catholic Kwandong University. Written informed consent was obtained from individual patients for the use of their tissue samples. Liposuction was performed under general anesthesia and sterile conditions, and the adipose tissue samples were harvested from the abdominal wall using gentle manual techniques. The ADASs were isolated from adipose tissues using a SmartX kit (Dongkoo Bio & Pharma Co., Seoul, South Korea) following the manufacturer’s instructions. After plating on culture dishes, non-adherent cells were discarded by changing the culture medium, and ADASs were passaged 4 times. The ADASs and ASCs (Lonza, Walkerville, MD, USA) were cultured in 10% fetal bovine serum (FBS; HyClone, Logan, UT, USA)-supplemented Dulbecco’s modified Eagle’s medium (DMEM; HyClone, Logan, UT, USA) and 1% penicillin/streptomycin at a density of 5×10^4^ cells/cm^2^ in a 100-mm dish in a humidified atmosphere with 5% CO_2_ at 37 °C. The ASCs were passaged 4 times.

### 2.2. Preparation of Normoxic and Hypoxic Conditioned Cells

The ASCs and ADASs were incubated with serum-free media (SFM) under normoxic or hypoxic conditions for 6, 12 or 24 h. For the hypoxic conditions, cells were incubated at 37 °C in 5% CO_2_, 5% H_2_ and 0.5% O_2_ in a chamber with an anaerobic atmosphere system (Technomart, Seoul, Korea). The cells were harvested after the 6-, 12- or 24-h incubation period and treated with RIPA buffer (Cell Signaling Technology, Danvers, MA, USA) for immunoblot analysis and with TRIzol Reagent (Life Technologies, Frederick, MD, USA) for quantitative real-time RT-PCR (qRT-PCR) analysis.

### 2.3. Flow Cytometry

For analysis of the surface markers of ADASs and ASCs, cells were incubated for 1 h with mouse IgG or rabbit IgG antibodies (against CD14, CD31, CD34, CD45, CD71, CD73, CD90, CD105, and CD106; Santa Cruz Biotechnology, Santa Cruz, CA, USA) on ice. Then, the cells were washed 3 times with phosphate buffered saline (PBS) containing 2.5% FBS and 0.1% sodium azide (washing solution). The cells were further incubated for 1 h with fluorescein isothiocyanate (FITC)-conjugated anti-mouse or rabbit IgG (Santa Cruz Biotechnology, Santa Cruz, CA, USA) on ice in the dark. Then, the cells were washed 3 times with washing solution and analyzed via flow cytometry using a BD Accuri^TM^ C6 Cytometry system (BD Bioscience, Piscataway, NJ, USA).

### 2.4. RNA Isolation, Reverse Transcription (RT)-PCR and Quantitive PCR (qPCR) Analysis

Total RNA was isolated from ASCs and ADASs using TRIzol Reagent and cDNA was synthesized using a Maxime RT PreMix kit (iNtRON Biotechnology, Seongnam, Korea). The level of each gene transcript was analyzed using an Applied Biosystems StepOnePlus real-time PCR system (Foster City, CA, USA) with SYBR Green Dye (TaKaRa Bio, Inc. Foster City, CA, USA). All values are shown as the normalized target gene expression level (fold change; 2^∆∆Ct^) by the glyceraldehyde 3-phosphate dehydrogenase (GAPDH) transcript level. All primers were designed using Primer3 and Basic Local Alignment Search Tool (BLAST) (Table 1).

### 2.5. Immunoblot Analysis

In the ADASs and ASCs, the levels of cardiomyogenic markers were investigated by immunoblot analysis according to the methods outlined in our previous studies [20,21,22]. Briefly, cell lysates were prepared with RIPA buffer containing 1% phosphatase inhibitors (Sigma-Aldrich, St. Louis, MO, USA), 1% protease inhibitors (Sigma-Aldrich, St. Louis, MO, USA) and 1% proteasome inhibitors (MG132; Abcam, Cambridge, UK), and 20 μg of extract per lane was used for SDS-PAGE. The transferred polyvinylidene difluoride (PVDF; Sigma-Aldrich, St. Louis, MO, USA) membranes were successively incubated with the appropriate primary antibodies and horseradish peroxidase (HRP)-conjugated secondary antibodies (Santa Cruz Biotechnology, Santa Cruz, CA, USA), and they were visualized by an enhanced chemiluminescence (ECL, GE Healthcare, Buckinghamshire, UK) system.

### 2.6. Immunocytochemistry

The cells were cultured in 4-well slide chambers (SPL, Pocheon, Korea) at a density of 2×10^4^ cells/well and permeabilized using 0.1% Triton X-100 (Sigma-Aldrich) for 10 min. Then, the samples were blocked for 1 h in a blocking solution (2.5% normal horse serum (Vector Laboratories, Burlingame, CA, USA) in PBS) and incubated with mouse anti-troponin T (Abcam, Cambridge, UK) and rabbit anti-GATA binding protein-4 (GATA-4; Abcam, Cambridge, UK) antibodies, diluted at 1:500, overnight at 4 °C. Rhodamine-conjugated goat anti-mouse IgG (EMD Millipore, Burlington, MA, USA) and FITC-conjugated goat anti-rabbit IgG (Vector Laboratories) were used as secondary antibodies, diluted at 1:500. The nuclei were stained with 4′,6-diamidino-2-phenylindole (DAPI; Invitrogen, Carlsbad, CA, USA), and immunofluorescence was detected by a confocal microscope (LSM710; Carl Zeiss Microscopy GmbH, Jena, Germany).

### 2.7. Antibody Array

Secretome analysis was performed in CM harvested from ASCs and representative ADASs under normoxic or hypoxic conditions using proteome profiler antibody arrays. Secretory profiles were determined with the Human XL Cytokine Array Kit (R&D Systems (ARY022B), Minneapolis, MN, USA) following the manufacturer’s instructions.

### 2.8. Statistical Analysis

All data were compared via one-way analysis of variance (ANOVA), which is a parametric test, using the Statistical Package of Social Science (SPSS, version 14.0K; SPSS Inc., Chicago, IL, USA) program. In our study, for comparing multiple groups, ANOVA was performed followed by Bonferroni tests. The quantified data are the averages of at least triplicate samples, and the error bars represent the SD of the mean. The *p* values ˂0.05 were considered significant based on the protected least-significant difference (LSD) test.

## 3. Results

### 3.1. Characterization of ADASs

To investigate the differentiation-related changes in ADASs by the time of hypoxic stress, we obtained ADASs from four donors (Figure 1A). Isolated ADASs were cultured until passage 4 to obtain sufficient numbers for hypoxic conditions, and we observed the surface protein expression of the ASCs using flow cytometry. The ADASs and ASCs were positive for CD14 (14.8% vs. 37.1%), CD73 (90.7% vs. 84.7%), CD90 (94.2% vs. 96.1%) and CD105 (88.9% vs. 93.1%) and negative for CD31, CD34, CD45, CD71 and CD106 (Figure 1B), showing that the ADASs used in this study contain ASCs.

### 3.2. Time-Dependent Differential Regulation of Differentiation-Related Genes of Cell Lineages in ADASs under Hypoxic Conditions

To study the differentiation-related changes in ADASs according to the time under hypoxic conditions, we cultured the ADASs under normoxic or hypoxic conditions for 6, 12 or 24 h, and hypoxic genes, as well as adipocyte (*PPARG, LPL, FABP4*), chondrocyte (*SOX9, ACAN, COL2A1*), osteoblast (*COL1A1, RUNX2, OCN, ALPL, COL2A1*) and cardiomyocyte markers (*GATA4, TBX5, NKX2.5*), were investigated using qRT-PCR analysis (Figure 1A). The ASCs were used as a control because ADASs are a multi-cell population that contains ASCs. Cardiomyogenic gene expression was significantly increased by hypoxic stress in ADASs, but not ASCs, whereas adipocytes, chondrocytes, and osteoblast differentiation-related genes showed no differences between the normoxic and hypoxic conditions in both ASCs and ADASs (Figure 2).

### 3.3. Time-Dependent Differential Regulation of Cardiomyogenic Proteins in ADASs under Hypoxic Conditions

In Figure 2, we found that cardiomyogenic gene expression was remarkably enhanced by hypoxic stress in only ADASs in a time-dependent manner. In addition to differentiation-related genes, more cardiomyogenic proteins, such as troponin T, MyoD, myosin heavy chain (MHC) and caveolin-1, with GATA-4, TBX5, and NKX2.5 were investigated in ASCs and ADASs under normoxic or hypoxic conditions by immunoblot analysis (Figure 3). Although there were variations in individual ADASs and hypoxic time, TBX5 showed an increasing tendency in hypoxic stress in both ASCs and ADASs, and GATA-4, NKX2.5, troponin T, and myosin HC exhibited incremental changes following hypoxic stimulation in ADAS, whereas they showed a tendency to decrease in ASCs (Figure 3). However, MyoD and caveolin-1 were unaffected by hypoxic conditions in both ADASs and ASCs because they showed ragged and uneven expression under hypoxic stress (Figure 3). In addition, the expression of cardiomyogenic markers, such as GATA-4 and troponin T, in ASCs and ADASs in response to 24 h of hypoxia was also determined by immunocytochemical staining, and these markers showed the same results with immunoblot analysis (Figure 4).

### 3.4. Changes in Cardiomyogenic Markers in Dimethyloxaloylglycine -Treated ADASs

To examine the association of HIF-1α with the increase in cardiomyogenic factor expression in ADASs following hypoxic stress, we added dimethyloxaloylglycine (DMOG), which leads to HIF-1 activation under nonhypoxic conditions, to ASCs and ADASs, and cardiomyogenic markers were investigated by qRT-PCR and immunoblot analysis (Figure 5). Though hypoxic stress time which showed highest expression levels of cardiomyogenic factors was different according to individuals and factors (Figure 2 and Figure 3), majority of individuals and factors exhibited highest expression levels under the hypoxic condition for 12 h. Therefore, we treated DMOG to cells for 12 h. As expected, the gene and protein expression of HIF-1α was induced by DMOG treatment in both ASCs and ADASs, and cardiomyogenic markers, such as TBX5, NKX2.5, troponin T, and myosin HC, were significantly increased by DMOG treatment in only ADAS, although the time point of the peak gene and protein levels was different according to individuals (Figure 5). These results suggest that cardiomyogenic factors induced by hypoxic stress in ADASs were increased in association with the expression of HIF-1α.

### 3.5. Secretome Analysis of ASCs and ADASs under Hypoxic Conditions

To identify causative factors that induce the expression of cardiomyogenic markers in only ADASs by hypoxic stress within 24 h, we analyzed the secretomes in the CM of ASCs and representative ADASs under normoxic or hypoxic conditions using an antibody array with a Proteome Profiler Human Cytokine Array kit (Figure 6). A total of 105 different cytokines, chemokines and acute phase proteins (Appendix A) were investigated in the CM of ASCs and ADASs under normoxic or hypoxic conditions using an antibody array. Among secreted proteins, proteins increased by hypoxic stress in ADASs and proteins with higher levels in ADASs than ASCs under hypoxic conditions were selected. We identified three proteins: angiogenin, fibroblast growth factor-19 (FGF-19) and macrophage inhibitory factor (MIF).

## 4. Discussion

MSCs have advantages for cellular therapy, as they are safe, multipotent and immune-privileged [23], but ASCs are more preferable cell sources due to their merits, such as easy accessibility, minimal morbidity upon harvest, clinically relevant levels and strong proliferative potential [24]. In contrast, SVF is an aqueous fraction derived from the enzymatic digestion of lipoaspirate by liposuction and includes various cell populations and secreted factors [3]. ADASs are cells without non-adherent cells and secreted factors in SVF and include some cell types containing ASCs [6]. SVF treatment was shown to have therapeutic effects similar to those of ASC treatment on osteochondral defects and myocardial infarction [25,26].

Human ASCs reside in a microenvironment with low oxygen tension in the body and experience various levels of oxygen tension (1–5% O_2_) [27]. These cells adapt themselves to hypoxic microenvironments by regulating their proliferation, differentiation, and other physiological processes [28]. Some studies have demonstrated that hypoxia influences proliferation, differentiation and cytokine or growth factor secretion in human ASCs [27,29]. In particular, hypoxia may influence the differentiation of smooth muscle cells [30], adipocytes [31], endothelial cells [32], osteocytes, chondrocytes [33] and myocardial-like cells [34] from ASCs. Therefore, hypoxic preconditioning was considered to be an effective method for MSC therapy [35]. Actually, hypoxic preconditioning may increase cell viability, retention of transplant, angiogenesis, and modulation of angiogenic factors such as vascular endothelial growth factor (VEGF) and interleukin (IL)-6 following transplantation in human MSCs [36,37,38], and a conditioned medium from hypoxic preconditioning can lead MSC differentiation into myocardial-like cells [34]. In addition, implantation of hypoxic ASCs promoted therapeutic angiogenesis and cardiac function recovery in the infarcted myocardium [39]. Accordingly, hypoxic conditions prior to ASCs implantation into patients may be an available strategy for effective implantation [2,40].

We previously observed that BM-MSCs altered their expression of several cardiomyocyte differentiation-related factors following hypoxic stress [18]. Therefore, we hypothesized that hypoxia may also influence the differentiation of isolated ADASs from SVFs and ASCs into cardiomyocytes and/or cells with comparable phenotypes. This phenomenon is because the ASCs have similar characteristics to BM-MSCs [4,41]. The reason why we used ADASs but not SVF is that ADASs are can be stored and used when necessary. First, we investigated the surface proteins of ADASs and ASCs using flow cytometry to compare two cell types (Figure 1B). The results were consistent with those of other studies [5,42] and suggested that ADASs contain ASCs (Figure 1B). Quantitative differences of surface proteins between ADASs and ASCs were attributed to the heterogeneity of ADASs (Figure 1B).

Based on these initial results, we hypothesized that ADASs and ASCs will show similar responses to hypoxic stress. However, in the present study, we observed that only ADASs expressed cardiomyogenic differentiation markers within 24 h under hypoxic conditions (Figure 2, Figure 3 and Figure 4). ASCs may differentiate into VECs, VSMCs, and cardiomyocytes, which are major components of the cardiovascular system, in vitro and in vivo but may fuse with tissue-resident cells and obtain their characteristics [2]. In other words, ASCs may express markers after fusion with VECs, VSMCs, and cardiomyocytes without direct differentiation [2]. In this regard, the cardiomyogenic factors showed more rapid expression in ADASs than ASCs under hypoxic conditions (Figure 2, Figure 3 and Figure 4) because ADASs are a mixed cell population. We also found that cardiomyogenic genes and proteins were induced in hypoxic ADASs in association with the expression of HIF-1α (Figure 5) which is a key mediator of responses to low oxygen tension [43]. In other words, hypoxia-induced signaling controls cardiac development through the HIF-1α-mediated transcriptional regulation of ADASs, thereby providing a mechanistic basis of how heart development is coupled to low oxygen tensions.

We performed an antibody array to identify causative factors that induce the expression of cardiomyogenic markers in only ADASs by hypoxic stress and found the increased expression of three proteins—angiogenin, FGF-19, and MIF—among 105 different proteins (Appendix A) in ADASs compared with ASCs under hypoxic conditions (Figure 6). Angiogenin is a strong stimulator of new blood vessels through angiogenesis and interacts with endothelial and smooth muscle cells, resulting in cell migration and proliferation [43]. In vitro studies on isolated cardiomyocytes demonstrated that angiogenin may inhibit the translation of stress-related proteins through the promotion of cardiomyocyte survival [44]. In addition, Huang et al. observed the effects of transplanted autologous MSCs overexpressing angiogenin on myocardial perfusion and cardiac function in a porcine chronic ischemic model [45]. However, there is no report about the association between angiogenin and cardiomyocyte differentiation to our knowledge. Most FGFs bind cell surface tyrosine kinase FGF receptors (FGFRs) and then function in a paracrine/autocrine manner to induce cell proliferation and differentiation [46,47,48]. A recent study observed that plasma FGF-19 levels in patients with coronary artery disease (CAD) were reduced in the Chinese population and negatively related to the severity of CAD [49]. However, studies that have assessed the association between FGF-19 and cardiovascular disease are limited. MIF participates in the pathogenesis of various inflammatory diseases as a proinflammatory factor and is produced and stored in various cell types, such as immune cells, endothelial cells and cardiomyocytes [50,51]. In particular, this molecule is rapidly released from intracellular stores by various stimuli, including infection, inflammation, and hypoxia [51]. MIF was shown to reduce the infarct size and preserve cardiac function by activating AMP-activated protein kinase (AMPK), inhibiting c-Jun N-terminal kinase (JNK)-mediated apoptosis and decreasing cardiomyocyte oxidative stress in the ischemic heart [52,53]. However, whether MIF protects the injured heart is unknown. In this manner, although there is no direct association between the three proteins and cardiomyocyte differentiation, several reports suggested they may contribute to cardiac diseases, such as the activation of cardiac stem cells, the promotion of their survival, and proliferation and/or differentiation into other cells.

## 5. Conclusions

Cardiomyogenic factors were expressed more rapidly in ADASs than in ASCs under hypoxic conditions through association with the expression of HIF-1α, and we hypothesized that angiogenin, FGF-19 and/or MIF are related. These data suggested that ADASs preconditioned by hypoxic stimulation are effective in cardiac disease. However, further mechanism studies of ADASs preconditioned by hypoxic stimulation are certainly necessary for use in cardiac disease.

## Figures and Tables

**Figure 1 jcm-08-01231-f001:**
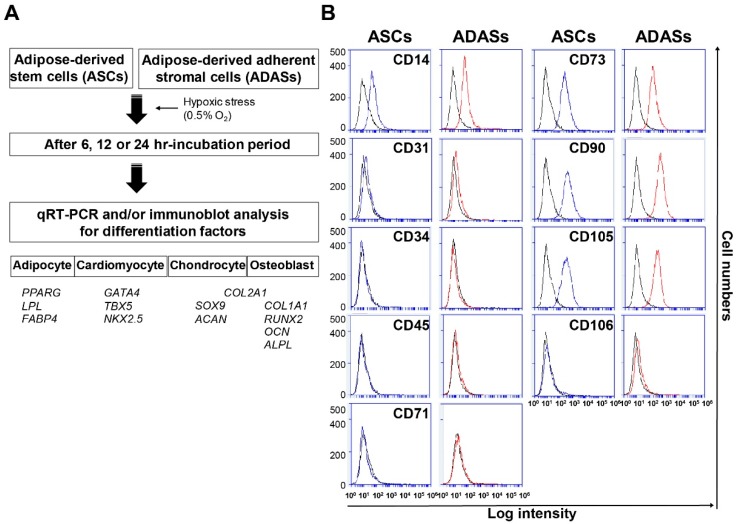
Experimental scheme of this study (**A**) and surface protein expression on adipose-derived adherent stromal cells (ADASs) and representative adipose-derived stem cells (ASCs), as measured by flow cytometry (**B**). Black histograms show the expression of isotype controls and blue and red histograms indicate the expression of target proteins on ASCs and ADASs, respectively. The data are representative of two independent experiments.

**Figure 2 jcm-08-01231-f002:**
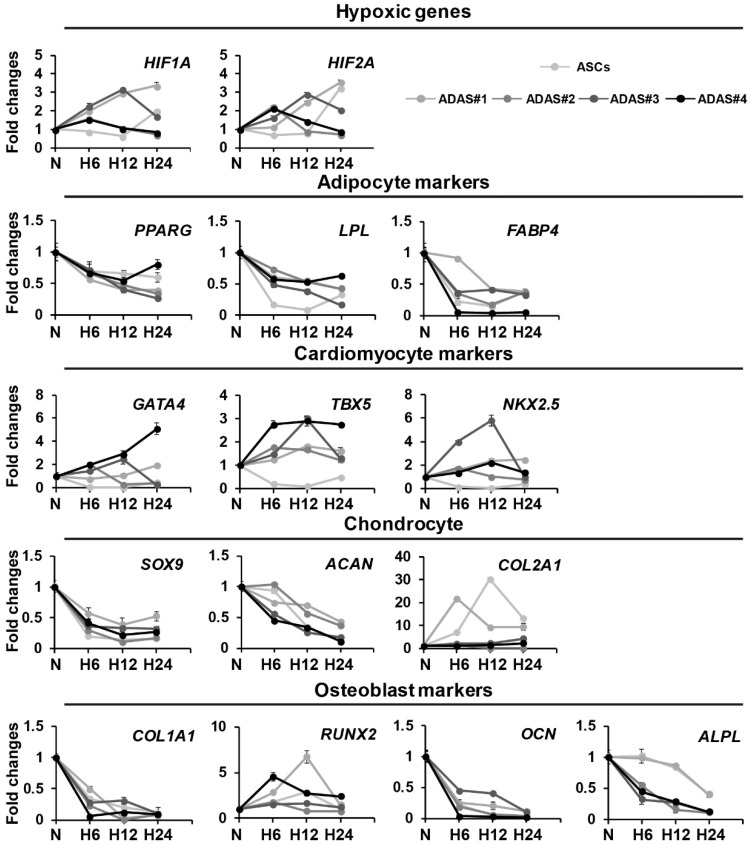
Time-dependent differential regulation of differentiation-related genes of cell lineages in ADASs under normoxic or hypoxic conditions as determined by qRT-PCR. The data are representative of three independent experiments. N, normoxia; H6, hypoxia for 6 h; H12, hypoxia for 12 h; H24, hypoxia for 24 h. Statistical significance between the normoxic and hypoxic groups is indicated in Appendix A.

**Figure 3 jcm-08-01231-f003:**
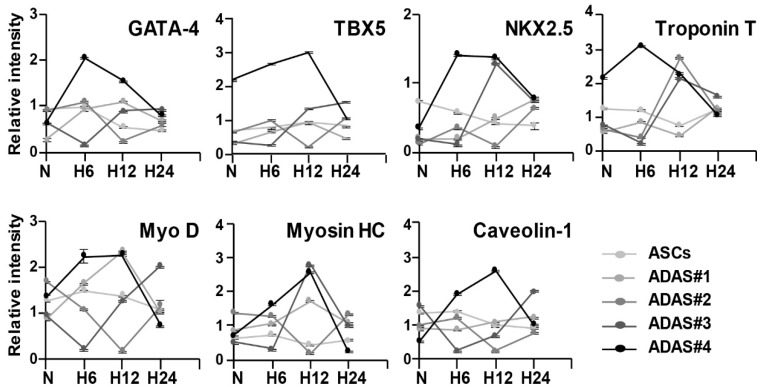
Time-dependent differential regulation of cardiomyogenic proteins in ASCs and ADASs under normoxic and hypoxic conditions as determined by immunoblot analysis. Band intensity was measured as area density and analyzed in ImageJ. Relative intensity levels indicate protein levels normalized to β-actin levels. The data are representative of two independent experiments. N, normoxia; H6, hypoxia for 6 h; H12, hypoxia for 12 h; H24, hypoxia for 24 h. Statistical significance between the normoxic and hypoxic groups is indicated in Appendix A. Western images are shown in Appendix A.

**Figure 4 jcm-08-01231-f004:**
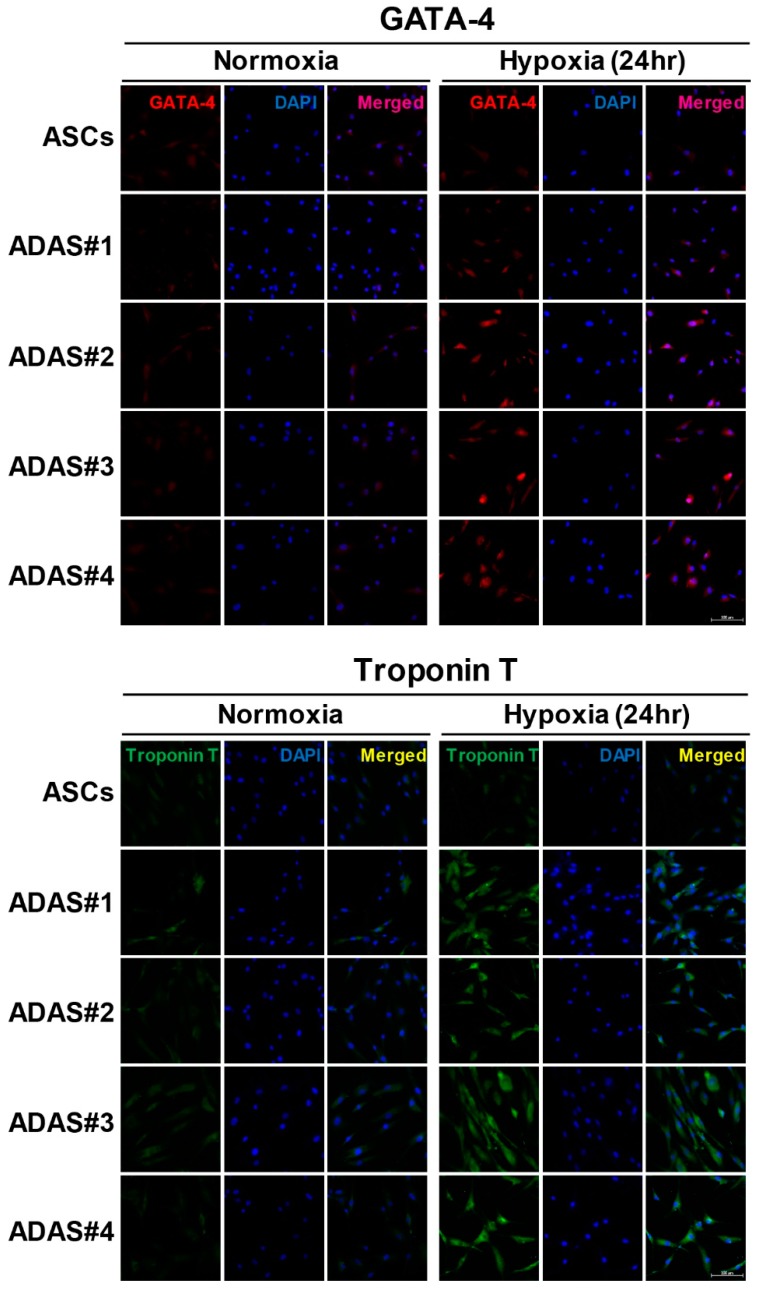
Expression of cardiomyogenic proteins, GATA binding protein-4 (GATA-4) and troponin T in ASCs and ADASs in response to hypoxia (24 h) as determined by immunocytochemical staining. Nuclei were stained with DAPI. Scale bar = 100 μm.

**Figure 5 jcm-08-01231-f005:**
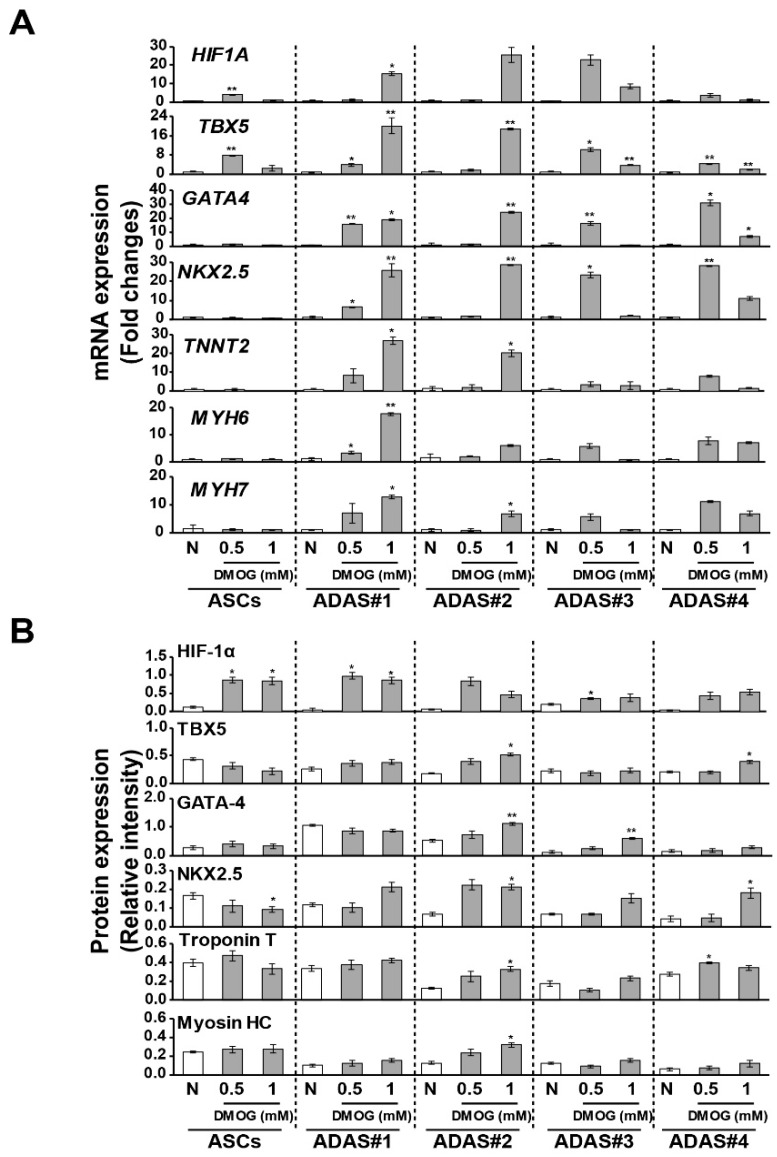
Changes in cardiomyogenic factors in dimethyloxaloylglycine (DMOG)-treated ASCs and ADASs as determined by qRT-PCR (**A**) and immunoblot analysis (**B**). All values of qRT-PCR are shown as the normalized target gene expression level relative to *GAPDH* transcript levels. *p* values indicated as * *p* < 0.05 and ** *p* < 0.01 between the control group and DMOG-treated group. The data are representative of two independent experiments. Immunoblot images are shown in the supplementary figure.

**Figure 6 jcm-08-01231-f006:**
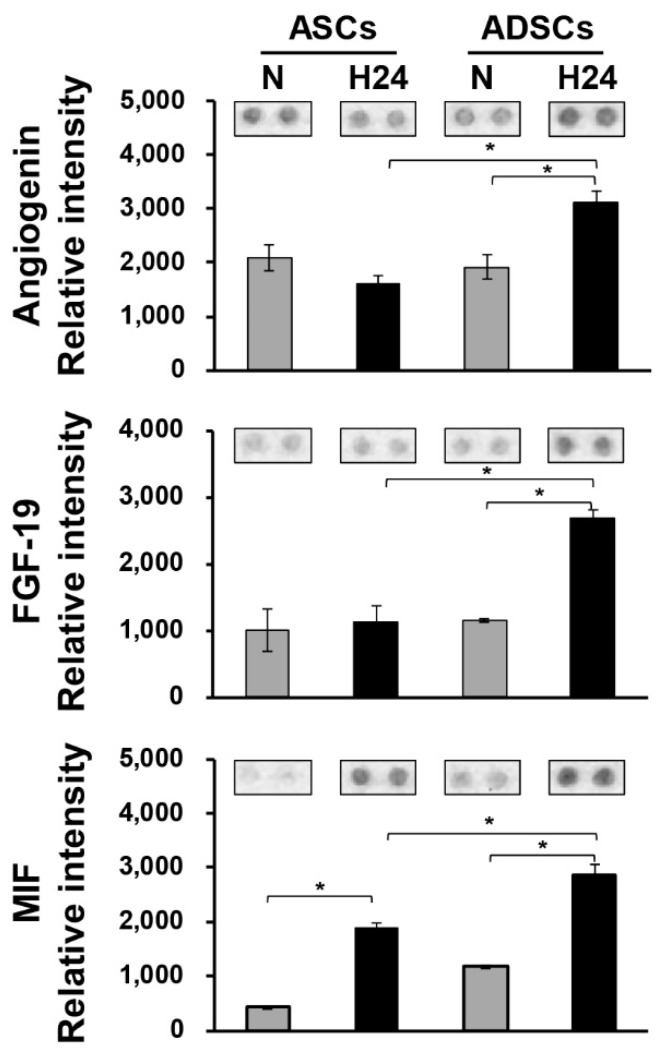
Secretome analysis in a conditioned medium (CM) of ASCs and representative ADASs under normoxic or hypoxic conditions. Statistical significance between hypoxic ASCs and hypoxic ADAS and between normoxic ADASs and hypoxic ADASs was determined by ANOVA using SPSS, with *p* values indicated as * *p* < 0.05. N, normoxia; H24 hypoxia for 24 h.

**Table 1 jcm-08-01231-t001:** Sequences of primers used for qPCR.

Genes		Primer Sequence (5′–3′)
*HIF1A*	F ^1^	ACTTGGCAACCTTGGATTGG
	R ^2^	GTGCTGAGTAACCACCACTTAT
*HIF2A*	F	CCTGAGACTGTATGGTCAGCTC
	R	AGGACGGAGAGAAGGGAACC
*PPARG*	F	GCAAACCCCTATTCCATGCTG
	R	ACGGAGCTGATCCCAAAGTT
*LPL*	F	CGAGCGCTCCATTCATCTCT
	R	CCAGATTGTTGCAGCGGTTC
*FABP4*	F	CCTTAGATGGGGGTGTCCTG
	R	AACGTCCCTTGGCTTATGCT
*GATA4*	F	GCCTCTCGGTGTGACGAGT
	R	TGGTTCCGGAAGCTGATGTAG
*TBX5*	F	AAGAGTTCCCTCCTCTCCCC
	R	GTCTTGGCCCCGGGAATAAA
*NKX2.5*	F	CAACATGACCCTGAGTCCCC
	R	TAATCGCCGCCACAAACTCT
*SOX9*	F	AACAACCCGTCTACACACAGCTCA
	R	TGGGTAATGCGCTTGGATAGGTCA
*ACAN*	F	GTGGTGATGATCTGGCACGAG
	R	CGTTTGTAGGTGGTGGCTGTG
*COL2A1*	F	TGGTCTTGGTGGAAACTTTGCTGC
	R	AGGTTCACCAGGTTCACCAGGATT
*COL1A1*	F	CCGGAAACAGACAAGCAACCCAAA
	R	AAAGGAGCAGAAAGGGCAGCATTG
*RUNX2*	F	AAGGGTCCACTCTGGCTTTG
	R	CTAGGCGCATTTCAGGTGCT
*OCN*	F	TCACACTCCTCGCCCTATT
	R	TGAAAGCCGATGTGGTCAG
*ALPL*	F	GACCCTTGACCCCCACAAT
	R	CGCCTCGTACTGCATGTCCCCT
*TNNT2*	F	GACAGAGCGGAAAAGTGGGA
	R	CACAGCTCCTTGGCCTTCTC
*MYH6*	F	ACCAACCTGTCCAAGTTCCG
*MYH7*	F	GACACACTTGAGTAGCCCAGG
	R	CTTCTAGCCGCTCCTTCTCTG
***Internal control***		
*GAPDH*	F	CATGGGTGTGAACCATGAGA
	R	GGTCATGAGTCCTTCCACGA

^1^ F, sequence of sense strands; ^2^ R, sequence of anti-sense strands.

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
