# Peer review of "Hypoxia Rapidly Induces the Expression of Cardiomyogenic Factors in Human Adipose-Derived Adherent Stromal Cells"

_jcm, 2019, doi:10.3390/jcm8081231_

Round 1
Reviewer 1 Report
The study performed by Jung-Won Choi, et al., focused on the " impact of Hypoxia on the Expression of Cardiomyogenic Factors in Human Adipose-Derived Adherent Stromal Cells.
I declare no financial relationships with any organisations that might have an interest in the submitted review. I claim no other links or activities that could appear to have influenced the proposed review.
The objective and the data of the study is compelling. However, the structure of the manuscript and the analysis of the data are inappropriate. The document is hard to read.
The authors must propose statistical tests appropriate to the data (parametric or non-parametric tests) of the repeated data (to evoke kinetics) or individual data but direct comparisons, all with corrections for multiple comparisons.
They must only speak of an association of expression, primarily when they refer to "HIF-1α-dependent mechanism".
Association is not causality. The authors must further discuss biological hypotheses. Otherwise, the association is only the result of chance.
Author Response
Responses to the reviewers’ comments
We would like to thank the editor and the reviewers again for their careful reading and valuable comments that have really helped us improve the current revised version of our manuscript. We are pleased that the referee considered our work important contribution to the field and worth publishing after some modifications. We have addressed all the general and specific comments made by the referee in order to improve the clarity of the manuscript. We agree with most of his/her propositions and updated our manuscript accordingly. Moreover, and in order to help the assessors in their reviewing process, we have marked all the new changes and corrections in the revised version in blue. A full point-by-point list of responses and explanations is provided below. This manuscript was edited for proper English language, grammar, punctuation, spelling, and overall style by one or more of the highly qualified native English speaking editors at American Journal Experts (Certificate Verification Key: 5LDD54DJ-B57E-2CCB-5503-69C3-CABP).
C: comments by the editor and reviewers; R: responses by the authors
Reviewer #1:
Comments for the authors:
The study performed by Jung-Won Choi, et al., focused on the "impact of Hypoxia on the Expression of Cardiomyogenic Factors in Human Adipose-Derived Adherent Stromal Cells.
I declare no financial relationships with any organisations that might have an interest in the submitted review. I claim no other links or activities that could appear to have influenced the proposed review.
[C1] The objective and the data of the study is compelling. However, the structure of the manuscript and the analysis of the data are inappropriate. The document is hard to read.
[R1] In the present study, we tried to find out effective preconditioning method for ADASs implantation and found rapid expression of cardiomyogenic factors in ADASs by hypoxic stress. For a reader to make easy to read, redundant explanations and unnatural expressions were deleted throughout the revised manuscript.
[C2] The authors must propose statistical tests appropriate to the data (parametric or non-parametric tests) of the repeated data (to evoke kinetics) or individual data but direct comparisons, all with corrections for multiple comparisons.
[R2] We thank for the cautious comment and agree with reviewer’s suggestion. Therefore, we have performed statistical analysis of all data, repeatedly. All data were compared via one-way analysis of variance (ANOVA), which is parametric test, using the Statistical Package of Social Science (SPSS, version 14.0K) program. In our study, for comparing multiple groups, ANOVA followed by Bonferroni tests was performed. The quantified data are the averages of at least triplicate samples, and the error bars represent the SD of the mean. The p values ˂0.05 were considered significant based on the protected least-significant difference (LSD) test. This information has added in experimental section.
[C3] They must only speak of an association of expression, primarily when they refer to "HIF-1α-dependent mechanism". Association is not causality. The authors must further discuss biological hypotheses. Otherwise, the association is only the result of chance.
[R3] We thank for the careful comment and changed "HIF-1α-dependent mechanism" into "association with expression of HIF-1α" throughout the revised manuscript by reviewer’s suggestion.
Reviewer 2 Report
Dear authors,
Thank you for allowing me to read your study. I have the following comments:
- first phrase in the introduction - there are two sentences, but the connection between them is imperfect
- second phrase - is mostly redundant - everybody in the field knows some CVD disorders.
- third phrase - you begin disussing treatment of MI, but say that heart failure causes the death of myocardial tissue... MI leads to myocardial necrosis, which means death of the myocardial tissue, which may cause acute or chronic heart failure. Overall, this part of the introduction shows a lack of proper understanding of cardiovascular disorders.
- overall, the study seems well conducted; however, taking into account the high number of factors that were analyzed, the low number of positive results ( for ADA, 3 out of 105, well within the margin or error of the p value), and the low number of subjects, I am not sure that the positive results are not false positives (type I statistical error). Therefore, I would see this study as a negative one
Author Response
Responses to the reviewers’ comments
We would like to thank the editor and the reviewers again for their careful reading and valuable comments that have really helped us improve the current revised version of our manuscript. We are pleased that the referee considered our work important contribution to the field and worth publishing after some modifications. We have addressed all the general and specific comments made by the referee in order to improve the clarity of the manuscript. We agree with most of his/her propositions and updated our manuscript accordingly. Moreover, and in order to help the assessors in their reviewing process, we have marked all the new changes and corrections in the revised version in blue. A full point-by-point list of responses and explanations is provided the attached file.

Reviewer 3 Report
Choi and collaborators in this paper evaluate the effect of hypoxia on human Adipose-Derived Adherent Stromal Cells obtained from SVF, compared to Adipose derived Stem cells, focusing on the expression of cardiomyogenic factors. The topic if of interest since it could suggest the use of an alternative source of stem cells as a therapeutic strategy for cardiac diseases.
Overall, the quality of the paper is good but could be improved.
The introduction is coincise but complete.
The material and methods section needs some revision:
it is not clear why the authors incubated the cells with DMOG for 12 hours (this point should be addressed here or elsewhere in the paper) paragraphs 2.6 and 2.7 have the same title, but the second one is related to immuno-fluorescence in the paragraph 2.6, a list of the antibodies used should be provided the protocol for antibody array (paragraph 2.8) should be briefly reported.
In the results, the authors report a immunophenotypic characterization of ADASs compared to ASCs. As I can understand from the text, the results reported are representative of the analysis carried on one of the four ADASs cell populations used in the study. What about the other three? Since in many of the following experiments the four cell populations show a high grade of heterogeneity, is it true also for the surface protein expression? In the legend of the figure, please explain the meaning of the different line colors (I guess that one is the curve os the isotype control or of the unstained control, but you should explain).
Regarding the expression of differentiation-related genes, have the authors (here or in a previous work) ever assessed the differentiation abilities of these cells? In the figure 2, please add the legend on X-ass on every diagram.
The authors analysed by immunoblotting the expression of some cardiomyogenic proteins. It is not clear why they decided to analyse two of them also by immunofluorescence, since the results obtained is not described in detail and in this way it does not add anything to the immunoblotting analyse.
As already stated above, the authors should explain why they incubated the cells with DMOG for 12 hours rather than for for 6 or 24 hours as in the other experiments.
Regarding the secretome analysis, it is not clear if the results reported come from a single experiment or from more than one. The 24-horus time has been chosen for a specific reason?
The discussion is somehow confusing, because it is not always clear if the authors are speking of MSC, Asc or ADAS and how they are related. Moreover, sometimes the discussion describes the results rather than discussing them (the authors also cite some figures in the discussion).
The authors show that ADAS have a high heterogeneity. Could this fact iinfluence their possible therapeutic efficacy? Since as the authors states that ADASs include different cell types, among which ASCs, is it possible to estimate the ASC content in a specific ADAS preparation and does this parameter possibly correlate with their biological properties?
Please revise the sentence on lines 281-282.
In conclusion, the paper by Choi and collaborators addresses in an original way an interesting topic. Some minor revision are need to improve its quality.
Author Response
Responses to the reviewers’ comments (jcm-572146)
We would like to thank the editor and the reviewers again for their careful reading and valuable comments that have really helped us improve the current revised version of our manuscript. We are pleased that the referee considered our work important contribution to the field and worth publishing after some modifications. We have addressed all the general and specific comments made by the referee in order to improve the clarity of the manuscript. We agree with most of his/her propositions and updated our manuscript accordingly. Moreover, and in order to help the assessors in their reviewing process, we have marked all the new changes and corrections in the revised version in blue. A full point-by-point list of responses and explanations is provided an attached file.

Round 2
Reviewer 2 Report
The authors have corrected the manuscript, and as it is, is acceptable for publication.